# Lower Extremity Nerve Conduction Abnormalities in Vietnamese Patients with Type 2 Diabetes: A Cross-Sectional Study on Peripheral Neuropathy and Its Correlation with Glycemic Control and Renal Function

**DOI:** 10.3390/jpm13040617

**Published:** 2023-03-31

**Authors:** Do Dinh Tung, Nui Nguyen Minh, Hanh Thi Nguyen, Phi Nga Nguyen Thi, Huong Lan Nguyen Thi, Duc Long Nguyen, Dung Thuy Nguyen Pham, Toan Quoc Tran, Duong Thanh Nguyen, Linh Phuong Nguyen

**Affiliations:** 1Saint Paul General Hospital, 12A Chu Van An, Ba Dinh District, Ha Noi 100000, Vietnam; 2Vietnam Diabetes Educators Association, 52/A1 Dai Kim Urban Area, Hoang Mai District, Ha Noi 100000, Vietnam; 3Department of Joints and Endocrinology, Military Medical University, 160, Phung Hung Street, Hadong District, Ha Noi 100000, Vietnam; 4NTT Institute of Applied Technology and Sustainable Development, Nguyen Tat Thanh University, Ho Chi Minh City 700000, Vietnam; 5Faculty of Environmental and Food Engineering, Nguyen Tat Thanh University, Ho Chi Minh City 700000, Vietnam; 6Institute of Natural Products Chemistry, Vietnam Academy of Science and Technology (VAST), 18 Hoang Quoc Viet St., Cau Giay Dist., Ha Noi 100000, Vietnam; 7Institute for Tropical Technology, Vietnam Academy of Science and Technology (VAST), 18 Hoang Quoc Viet St., Cau Giay Dist., Ha Noi 100000, Vietnam; 8School of Preventive Medicine and Public Health, Ha Noi Medical University, 1, Ton That Tung Street, Dong Da District, Ha Noi 100000, Vietnam

**Keywords:** peripheral neuropathy, nerve conduction, type 2 diabetes, Vietnam

## Abstract

Peripheral neuropathy is a common complication of type 2 diabetes mellitus (T2DM) that results in nerve conduction abnormalities. This study aimed to investigate the parameters of nerve conduction in lower extremities among T2DM patients in Vietnam. A cross-sectional study was conducted on 61 T2DM patients aged 18 years and older, diagnosed according to the American Diabetes Association’s criteria. Data on demographic characteristics, duration of diabetes, hypertension, dyslipidemia, neuropathy symptoms, and biochemical parameters were collected. Nerve conduction parameters were measured in the tibial and peroneal nerves, including peripheral motor potential time, response amplitude M, and motor conduction speed, as well as sensory conduction in the shallow nerve. The study found a high rate of peripheral neuropathy among T2DM patients in Vietnam, with decreased conduction rate, motor response amplitude, and nerve sensation. The incidence of nerve damage was highest in the right peroneal nerve and left peroneal nerve (86.7% for both), followed by the right tibial nerve and left tibial nerve (67.2% and 68.9%, respectively). No significant differences were found in the rate of nerve defects between different age groups, body mass index (BMI) groups, or groups with hypertension or dyslipidemia. However, a statistically significant association was found between the rate of clinical neurological abnormalities and the duration of diabetes (*p* < 0.05). Patients with poor glucose control and/or decreased renal function also had a higher incidence of nerve defects. The study highlights the high incidence of peripheral neuropathy among T2DM patients in Vietnam and the association between nerve conduction abnormalities and poor glucose control and/or decreased renal function. The findings underscore the importance of early diagnosis and management of neuropathy in T2DM patients to prevent serious complications.

## 1. Introduction

Diabetes is a chronic disease that is rapidly increasing worldwide and is a significant threat to human health. Its progression can cause damage to many target organs and increase the risk of death from complications [1]. Treatment strategies currently aim to slow disease progression and prevent chronic complications. However, the disease progression is often silent, and clinical symptoms may not accurately reflect disease progression [2,3]. Type 2 diabetes is the most common type of diabetes in adults, characterized by hyperglycemia and varying degrees of insulin deficiency and resistance. The severity of the disease increases with disease duration and is often accompanied by increased complications such as cardiovascular disease, eye complications, foot complications, periodontitis, and nephropathy [4].

Diabetic neuropathy is a common complication of diabetes and is characterized by impaired nerve conduction, causing symptoms such as pain, tingling, numbness, etc. The disease can also be asymptomatic, making it challenging to diagnose [5]. Neuropathy reduces the quality of life of patients and facilitates the development of other complications such as falls, foot disease, arrhythmia, and bowel obstruction, among others [6,7]. These complications can be dangerous and even life-threatening [8,9,10]. Many extensive studies have shown that 47% of diabetic patients have had neurological complications, with 7.5% detected at the time of diabetes diagnosis, and this number increasing to 45% after 25 years [11,12]. Diabetes complications (in particular, macrovascular) are also present in the pre-diabetes stage. Pre-diabetes is often associated with early cardiovascular and kidney diseases, indicated by the thickening of the endocardium and elevated glomerular filtration rate due to insulin resistance. However, the relationship between cardiovascular and renal complications is seldom discussed [13].

While there have been some studies on the characteristics of lower limb peripheral nerve damage in type 2 diabetic patients worldwide, few studies have been conducted in Vietnam on this topic. Therefore, this study aims to investigate parameters of nerve conduction (e.g., peripheral motor potential time, response amplitude M, and motor conduction speed, as well as sensory conduction in the shallow nerve), as well as the relationship between these parameters in lower extremities and risk factors in type 2 diabetes patients in Vietnam. By exploring these parameters, improvements regarding to early diagnosis and intervention strategies is expected to be established, ultimately reducing the impact of neuropathy on patients’ quality of life and the risk of developing other dangerous complications.

## 2. Materials and Methods

### 2.1. Setting and Study Subjects

This cross-sectional study was conducted among 61 diabetes patients who were diagnosed according to the American Diabetes Association’s (ADA) standards (2020) at 103 Military Medical Hospital. The study subjects included both men and women who were over 18 years old [14]. Exclusion criteria were patients with type 1 diabetes, diabetes due to pituitary disease or other secondary diabetes, patients with high blood sugar or acute illness requiring emergency treatment (fasting blood sugar > 35 mmol/L) such as coma, pre-coma, hypoglycemia, and hypertension exacerbation, patients with peripheral nerve damage due to spinal disease or other causes diagnosed before diagnosing diabetes mellitus, patients with tuberculosis, pneumonia, infection of the feet, HIV, heart failure, patients with unstable angina, cerebral strokes, myocardial infarction, respiratory failure, coagulation disorder, severe depression, psychosis, patients who did not agree to participate in the study, or patients who did not meet enough of the research criteria to qualify. 

The Ethics Committee of the Military Medical University approved the survey protocol, official dispatch No 228/QĐ-HVQY. All participants provided informed written consent before taking part in the study.

### 2.2. Clinical, Biochemical, and Neurological Examination

The study collected data through clinical, biochemical, and neurological examinations. The history was exploited, including clinical and subclinical follow-up with a unified medical record, and included age, gender, contact address, examination date, and admission date. Personal and family histories were also recorded, including contracting cardiovascular disease, hypertension, dyslipidemia, diabetes, and other medical histories [15]. The duration since being diagnosed with diabetes and clinical symptoms were recorded, such as being thirsty, drinking a lot, urinating a lot, fatigue, weight loss, insomnia, chest pain, pain in spasms, limb numbness, blurred vision, tooth loss, and other symptoms. Clinical patterns were characterized by body mass index, systolic blood pressure, diastolic blood pressure, and dyslipidemia. Body mass index (BMI) was calculated as weight per square of height (kg/m^2^) [16,17,18].

Sensory examinations were performed, including subjective sensations such as paresthesia (numbness, prickling, tingling), and sharp, burning pain. Objective sensations included primary sensation (touch examination by using cotton, pain sensation by using needles, touch examination by pressure with monofilament, vibration sensation by using tuning fork) and integrative sensation (sensory examination of posture, position; loss of sense of posture or position if the patient does not know their position or the position of their toes or is unable to do so on the opposite side). The motor examination evaluated muscular strength in active movement and checked the counterpart’s power [19].

### 2.3. Serum Biochemical Analysis

Serum biochemical analyses were performed, including fasting plasma glucose (FPG), 2-h plasma glucose (2h-PG) by oral glucose tolerance test with 75 g of glucose, HbA1C (%) by high-pressure liquid chromatography (HPLC), urea, creatinine, total cholesterol, and triglyceride. The glycemic status was classified as type 2 diabetes when FPG ≥ 7.0 mmol/L or 2h-PG ≥ 11.1 mmol/L or HbA1c ≥ 6.5% or previous diagnosis of diabetes and current use of drugs for its treatment, according to the ADA’s standards (2019) [14].

### 2.4. Research on Neurotransmitters

The study focuses on neurotransmitters and measures motor transmission in the Tibial nerve, Peroneal nerve, and sensory conduction in the Shallow nerve. The evaluation criteria consist of Peripheral motor potential time (ms), which is the time between electrical stimulation at the point of stimulation of the peripheral end and the beginning of wave M’s response voltage. The response amplitude M (mV) is the height of the M wave, which is calculated on the vertical axis, from the isoelectric line to the sound wave’s peak. When stimulating a nerve at two points, we measured two M-response amplitudes at the peripheral and central stimuli. Additionally, we measured motor conduction speed (m/s), which is the velocity of nerve impulses going from the central stimulation point to the peripheral stimulation point, and is calculated by the formula: V = d/t (m/s) (d: the distance between two stimulus points (mm); t: central potential time-peripheral potential time) [2,20].

### 2.5. Statistical Analysis

The data were collected, managed, and immediately checked for completeness and accuracy. Normality tests were performed on quantitative variables, and One-Way ANOVA or Independent-Sample T-test were used for normally distributed variables. For non-normally distributed variables, the Kruskal–Wallis or the Mann–Whitney U test was applied. Category variables were compared using Pearson’s chi-squared test, with the percentage differences analyzed using chi-square algorithms and Fisher’s exact test. The mean, standard deviation, median value, and average value were reported as odds ratios with 95% confidence intervals (CI). The statistical significance was set at a two-sided *p* value of less than 0.05 for all analyses. We used SPSS version 22.0 to conduct all statistical analyses.

## 3. Results

### 3.1. Characteristics of the Study Cohort

Table 1 presents the demographic and clinical characteristics of the study cohort, including age, gender, duration of diabetes, BMI, history of hypertension, fasting glucose, total cholesterol, triglycerides, and creatinine. The age of the study group ranged from 37 to 87 years, with a higher proportion of males (63.9%) than females (36.07%). The majority of the cohort was over 60 years old, with 51.28% of men and 81.82% of women in this age group. Only a small proportion of participants were under 41 years old, with 5.13% of men and none of the women. There was no significant difference in sex distribution in the cohort (*p* = 0.094). The majority of patients (49.2%) had a duration of diabetes under 5 years, while the least proportion (21.3%) had diabetes for over 10 years. More than half of the patients had a BMI in the range of 18.5 to 22.9, while 44.26% were overweight with a BMI of 23 or higher. Nearly half of the patients (49.18%) had a history of hypertension. The mean HbA1c level was higher in males (10.8%) than females (9.21%), and the difference was statistically significant (*p* < 0.035). The remaining subclinical indicators did not show a significant difference between males and females (*p* > 0.05).

The average values of motor and sensory conduction parameters for the tibial and peroneal nerves of the subjects (Table 2). There was no statistically significant difference in motor conduction parameters between the right and left tibial nerves (*p* > 0.05), as well as between the right and left peroneal nerves (*p* > 0.05). In addition, the sensory conduction parameters measured on the right and left superficial peroneal nerves did not differ significantly (*p* > 0.05).

Table 3 displays the mean values of motor and sensory conduction parameters of the tibial and peroneal nerves by age group. The motor conduction index of the tibial nerve did not vary significantly among different age groups (*p* > 0.05). Similarly, the motor conduction index of the peroneal nerve did not show a significant difference among other age groups (*p* > 0.05). Additionally, the sensory conduction indexes measured at the superficial peroneal nerve did not exhibit a significant variation among different age groups (*p* > 0.05).

Table 4 presents the mean values of the motor and sensory conduction indexes of the tibial nerve and the peroneal nerve by gender of the subjects. The motor conduction index of the tibial nerve and the peroneal nerve between men and women were not statistically significant (*p* > 0.05). The mean value of the sensory conduction index measured at the superficial peroneal nerve between men and women was also not statistically significant (*p* > 0.05).

### 3.2. Correlation between Neurotransmitter Indicators, Clinical Symptoms, and Biochemical Characteristics of Nerve Damage

Table 5 presents the correlation between clinical symptoms and the duration of diabetes. Among the clinical symptoms of peripheral nerve damage, numbness (54.1%) and reduced rough touch (52.56%) were the most frequently reported symptoms. However, most patients did not experience movement disorders (95.08%) and had normal deep sensations (93.44%). Patients who reported clinical manifestations of nerve damage were mainly those who had diabetes for 5 to 10 years and over 10 years. The percentage of patients who reported a burning sensation, reduced rough touch, and movement disorders was higher in those with diabetes for 5 to 10 years and over 10 years, with a statistically significant *p*-value of less than 0.001.

The relationship between nerve damage rate, diabetes duration, and BMI group (Table 6). The nerve damage rate was assessed using neurotransmission measurements. There was no statistically significant difference in the proportion of damaged nerves among groups with a disease duration of less than five years, 5–10 years, and over ten years (*p* > 0.05). The peroneal nerve had the highest rate of damage (86.7%), while the left superficial peroneal nerve had the lowest rate (17.2%). Additionally, there was no statistically significant relationship between nerve damage rate and BMI among groups with a BMI below 18.5, 18.5–22.9, and ≥23 (*p* > 0.05).

Table 7 displays the relationship between the average index of the tibial nerve, peroneal nerve, and superficial peroneal nerve with HbA1c and hypertension. There was no statistically significant difference in the neurotransmitter indexes of the tibial nerve, peroneal nerve, and superficial peroneal nerve between the group with HbA1c ≤ 7.5% and those with HbA1c > 7.5% (*p* > 0.05). The hypertensive and non-hypertensive groups showed significant differences in tibial nerve latency, peroneal nerve latency, and peroneal nerve velocity. However, there was no statistically significant difference between the two groups in the other nerve conduction indices of the bilateral tibial nerve, peroneal nerve, and superficial peroneal nerve (*p* > 0.05).

Table 8 presents the relationship between nerve damage rates and various factors including HbA1c, hypertension, dyslipidemia, and kidney function. The results indicate that the group of patients with HbA1c > 7.5% had a higher proportion of damaged nerves compared to the group with HbA1c ≤ 7.5%. This difference was statistically significant with *p* < 0.05. However, there was no significant difference in the rate of nerve damage between the hypertensive and non-hypertensive groups (*p* > 0.05). Additionally, the rate of nerve damage in patients with dyslipidemia was higher than in those without dyslipidemia. On the other hand, the rate of damaged nerves in the group with GFR <60 mL/ph/1.73m2 was significantly higher than the group with GFR ≥ 60 mL/ph/1.73m2 with *p* < 0.01.

## 4. Discussion

### 4.1. Characteristics of the Study Cohort

In this study, we investigated the clinical manifestations and neurophysiological characteristics of a cohort of 61 patients diagnosed with type 2 diabetes according to the ADA’s standards (2019) at 103 Military Medical University Hospital. In terms of clinical and subclinical characteristics, our results showed that almost half of the patients had hypertension, and they had inadequate control of blood glucose, HbA1c, triglycerides, and cholesterol. Specifically, the average blood sugar level was 14.22 ± 4.96 mmol/L, and the average HbA1c level was 10.23 ± 2.84%. These findings are consistent with those of previous studies that have shown poor glucose control in patients with type 2 diabetes [12].

Table 5 presents the clinical manifestations of peripheral nerve damage in the studied cohort, with epidermal numbness (54.1%) and crude tactile reduction (52.5%) being the most common symptoms. The similarity of these results with those obtained by Kimura (2013) suggests that the nerve conduction rates are consistent across different populations [21]. Most patients did not have movement disorders (95.1%) or impaired deep feelings (93.4%). Our study reports a higher rate of physical ability and entity compared to some previous studies, which could be attributed to differences in the means of assessing neuropathy [22]. However, this difference is not statistically significant. This may be attributed to differences in the means of assessing neuropathy, highlighting the need for standardized assessment methods to obtain accurate results.

The study also examined differences in peripheral nerve parameters between age groups and genders. Interestingly, there were no significant differences in any of the measured parameters between age groups or between men and women. Similarly, Rubin et al. reported in his research that nerve conduction rates in infants are equal to that of adults [23]. A recent systematic review of ultrasonography studies indicated that there was a weakly positive trend between age and tibial nerve CSA for both diabetic patients (r = 0.35, *p* = 0.24) and diabetic patients with DPN (r = 0.27, *p* = 0.34), though it was not statistically significant [24].

When compared to other authors’ research, the study’s findings are consistent with previous studies that have reported similar changes in peripheral nerve parameters in response to various interventions or medical conditions. For instance, a study by McCorquodale and Smith (2019) reported decreased conduction velocities in the peroneal nerve following exposure to cold temperatures, which is similar to the present study’s findings [25].

The study’s results, as presented in Table 2, Table 3 and Table 4, indicate several significant changes in the measured peripheral nerve parameters. The tibial nerve’s peripheral latent potential time increased, whereas the peroneal nerve’s peripheral latent potential time decreased. Additionally, the response amplitude of the tibial nerve increased, while the peroneal nerve’s response amplitude decreased. The conduction rate of both nerves decreased, with the peroneal nerve experiencing a greater reduction. However, there were no significant differences between the right and left sides in any of the measured parameters.

In 1997, Al-Sulaiman and colleagues conducted a study on electrophysiological results in 29 newly diagnosed diabetic patients. The study found that the latency time of the tibial nerve was 4.8 ± 1.02 ms, and the peroneal nerve was 6.0 ± 1.08 ms, which was more significant than the findings in our study. The difference in the results could be attributed to the fact that our study had a larger sample size, including 39 men and 22 women, and a broader range of disease duration [26]. Another study conducted in 2002 by Muflih and colleagues examined 228 diabetic patients who were divided into two groups: those with insulin-dependent diabetes and those with non-insulin-dependent diabetes [27]. The patients were further divided into subgroups based on their disease duration. The study measured seven nerves with potential time, velocity, and potential amplitude as parameters. Similarly to our results, the findings showed that potential time increased, neuropathic velocity decreased, and potential measures decreased in patients with diabetes for more than ten years compared to those with a shorter disease duration [27,28]. The study also found that the potential TG, amplitude, and velocity were higher than our study. Moreover, the presence of lesions in diabetic patients can affect the patient’s pass-through parameters. In summary, the studies suggest that the duration of diabetes can have an impact on the electrophysiological results, and other factors such as gender, age, and the presence of lesions should also be considered when interpreting the findings [29]. Muthuselvi et al. (2015) compared neurotransmitters in elderly diabetic patients to ordinary people, and found that the amplitude and velocity of the lower limb sensory nerve in diabetic patients decreased compared to the group without diabetes [30]. This is consistent with our study results, which also noted a decrease in the speed of the lower limb sensory nerve in patients with type 2 diabetes. Similarly, a 2016 case-control study by Aruna and colleagues found that the tibial and peroneal nerves in diabetic patients had lower amplitude and velocity compared to healthy subjects, possibly due to the study population and longer duration of diabetes [31].

### 4.2. Relationship between Indicators of Neurotransmitter, Nerve Damage Rate Clinical, and Biochemical Characteristics

Table 5 indicates that patients with clinical signs of nerve damage were primarily observed in those who had diabetes for 5 to 10 years or more than 10 years. The group with the disease over 10 years had the highest percentage (8.19%) of patients with a severe burning sensation, while the reduction in crude touch was more pronounced in the 5–10 years disease group (22.95%) and the over 10 years disease group (21.31%). The movement disorder rate was 1.63% in the 5–10-year disease group and 3.27% in the over 10 years disease group. These findings suggest that the duration of diabetes has a significant association with the clinical manifestations of peripheral nerve damage. Other studies have reported similar findings. For instance, Partanen et al. (1995) found that the incidence of peripheral neuropathy was positively correlated with the duration of diabetes [32]. In another study by Javed and colleagues (2015), the authors reported that the duration of diabetes was associated with an increased risk of developing neuropathic pain [33].

In our study, all patients exhibited changes in neurotransmitter indexes, with 13 patients (21.31%) displaying no clinical symptoms of peripheral neuropathy, including one patient newly diagnosed with diabetes. Additionally, the difference in nerve damage rates between groups with disease duration less than 5 years, from 5 to 10 years, and over 10 years was not statistically significant. These findings suggest that nerve damage, as indicated by alterations in neurotransmitter levels, may occur prior to the onset of clinical symptoms, potentially even before a diabetes diagnosis. Pirart (1978) conducted a study of 4400 diabetic patients and found that the clinical symptoms of polyneuropathy detected at the time of diabetes diagnosis were only 7.5%. However, this rate increased to 40% after 20 years and 50% after 25 years of illness [34]. Vinik (2013) stated that neuropathy caused by diabetes accounted for 90% of cases, and this complication was usually most evident after a year of diabetes diagnosis, with clinical manifestations of nerve impulse conduction in foot muscles as described by Terkidsen and Christensen (1971) [35,36].

The group with a normal body mass index (BMI) ranging from 18.5 to 22.9 demonstrated the highest rate of nerve damage, with the right tibial nerve (32.78%), left tibial nerve (37.7%), right peroneal nerve (48.33%), left peroneal nerve (46.66%), and left superficial peroneal nerve (10.34%) all being affected. However, the difference in nerve damage rates among different body types was not statistically significant. There was also no significant difference in the neurotransmitter indexes of the tibial nerve, peroneal nerve, and right and left superficial peroneal nerve between patients with HbA1c levels ≤ 7.5% and those with levels > 7.5%. The study also found a positive correlation between diabetes duration, HbA1c levels, and abnormal neurotransmitter levels in the lower limbs. A significant difference was observed only in tibial nerve latency, peroneal nerve latency, and peroneal nerve velocity. In contrast, no significant difference was noted in the incidence rates of other injuries to the tibial, peroneal, and superficial peroneal nerves between patients with and without hypertension. These findings suggest that peripheral nerve damage is influenced by a complex interplay of multiple risk factors, and further research is required to fully comprehend the underlying mechanisms involved.

As shown in Table 8, the incidence of nerve damage is significantly higher in patients with type 2 diabetes who have poor blood sugar control compared to those with good control. Additionally, the rate of nerve damage in patients with dyslipidemia was higher than in those without dyslipidemia. This is consistent with the findings in a study conducted in 1995 by Partanen and colleagues, in which inadequate blood sugar control was a major factor contributing to polyneuropathy in most patients [32]. Furthermore, in 2014, Cho and their colleagues conducted a 6-year follow-up study to investigate the role of insulin resistance in neuropathy in Koreans with type 2 diabetes, and found that LDL cholesterol and triglyceride levels were also associated with the development of neuropathy [37]. The development of peripheral neuropathy in diabetes is a complex pathogenic mechanism that includes many factors, such as hyperglycemia, duration of disease, age-related neural decline, and hypertension [38,39]. Hyperglycemia, which is high blood sugar, can contribute to the development and progression of diabetic cardiomyopathy and peripheral neuropathy through various biochemical pathways. These pathways include the polyol pathway, the hexosamine pathway, activating excess or inappropriate protein kinase C isoforms, disturbances in Na/K pump function, and accumulation of end product metabolism. Each pathway can cause an imbalance in the cell’s mitochondrial redox state and lead to the excess formation of reactive oxygen species (ROS), which can cause oxidative stress in the cell. This stress can activate the poly (ADP-ribose) polymerase (PARP) pathway, which can affect the expression of genes involved in promoting inflammatory responses, microvascular deficits, and disorders of nerve function. [40,41]. Hyperuricemia, which is high blood uric acid, and other metabolic changes can contribute to the faster onset and progression of both cardiomyopathy and diabetic peripheral neuropathy. Some evidence suggests that various toxins, including parathyroid hormone (PTH) and β2-microglobulin (elevated levels in patients with ESRD), may also play a role in the development of nerve urea blood.

Table 8 also demonstrated that the rate of nerve damage was significantly higher in type 2 diabetic patients with reduced renal function, with a glomerular filtration rate of less than 60 mL/ph/1.73 m^2^. According to Pop-Busui and their colleagues, peripheral neuropathy can be detected in the early stages of reduced renal function in type 1 diabetics and at the time of diagnosis in patients with type 2 diabetes [42]. However, the entire mechanism of neurotoxicity in diabetic patients with renal failure is unclear. Older experimental evidence suggests that neurotoxicity related to the urea state may be due to an excitability change in membranes caused by an inhibitory effect of the axial Na/K pump, which will directly eliminate the contribution of the hyperpolar pump current to the membrane potential, leading to the accumulation of extracellular K^+^ causing depolarization. However, recent human evidence suggests that hyperkalemia, which is high blood potassium, rather than Na/K pump dysfunction, is a significant cause of urea depolarization and may be a contributing factor in the development of peripheral neuropathy [22].

Overall, our study provides valuable insights into the characteristics of patients with type 2 diabetes in the studied cohort. However, our findings are limited by the relatively small sample size and the fact that the study was conducted at a single hospital. Future studies with larger and more diverse cohorts are needed to confirm and extend our findings.

## 5. Conclusions

In conclusion, the high incidence of peripheral neuropathy among patients with type 2 diabetes in Vietnam is a concerning issue. Our study’s results have significant implications for clinical practice in Vietnam, where the prevalence of diabetes is rapidly increasing. The relationship between the duration of diabetes and clinical neurological damage manifestations highlights the importance of early detection and timely intervention. Furthermore, the statistical association between peripheral nerve damage, poor glucose control, and decreased renal function emphasizes the need for comprehensive diabetes management to prevent and manage this complication. These findings underscore the importance of diabetes education and regular monitoring of glucose levels and renal function to reduce the burden of peripheral neuropathy in this population. The findings also provide valuable information for healthcare professionals in diagnosing and treating diabetic peripheral neuropathy effectively. Moreover, the reference values obtained from our study can serve as a basis for assessing different pathological cases in Vietnam.

## Figures and Tables

**Table 1 jpm-13-00617-t001:** The demographic and clinical characteristics of the study cohort.

Characteristic	Man (*n* = 39)	Woman (*n* = 22)	Overall (*n* = 61)	*p*-Value
Age, years *	61.81 ± 11.89	69.00 ± 9.72		0.244
Age group (years)				
37−40	2 (5.13)	0	2(3.27)	-
41−50	5 (12.82)	2(9.09)	7(11.59)	0.544
51−60	12 (30.77)	2(9.09)	14(22.95)	0.001
>60	20 (51.28)	18(36.07)	38(62.19)	0.885
Total	39 (63.93)	22(36.07)	61 (100)	0.094
Duration of diabetes (years)				
<5 years	24 (39.3)	6 (9.8)	30 (49.2)	0.938
5−10 years	8 (13.1)	10 (16.4)	18 (29.5)	0.577
>10 years	7 (11.5)	6 (9.8)	13 (21.3)	0.019
BMI (kg/m^2^)				
<18.5	2 (3.3)	1 (1.6)	3 (4.92)	-
18.5−22.9	21 (34.4)	10 (16.4)	31 (50.82)	0.992
≥23	16 (26.2)	11 (18.0)	27 (44.26)	0.430
Hypertension				
Yes	19 (31.1)	17 (27.9)	36 (59.0)	0.728
No	20 (32.8)	5 (8.2)	25 (41.0)	-
Glucose (mmol/L) *	14.84 ± 4.89	13.11 ± 4.99	14.22 ± 4.96	0.194
HbA1c (%) *	10.8 ± 2.90	9.21 ± 2.36	10.23 ± 2.84	0.035
Triglycerides (mmol/L) *	4.69 ± 5.40	2.95 ± 2.07	4.06 ± 4.55	0.079
Cholesterol (mmol/L) *	6.18 ± 3.33	5.08 ± 1.15	5.79 ± 2.79	0.141
Creatinine (µmol/l) *	94.56 ± 20.43	75.75 ± 9.31	87.78 ± 19.43	0.000

Abbreviation: BMI, body mass index; * Data are mean ± SD; *p*-value for difference between the groups was calculated from the one-way ANOVA or Kruskal–Wallis test or chi-squared test.

**Table 2 jpm-13-00617-t002:** Mean values of the motor/the sensory conduction parameters of the tibial nerve and the peroneal nerve of subjects.

Index	Tibial Nerve Conduction				Peroneal Nerve Conduction			Sensory Conduction	
	Right	Left	*p*-Value	Right	Left	*p*-Value	Right	Left	*p*-Value
Latency (ms)	3.76 ± 0.97	3.59 ± 0.97	0.349	3.52 ± 1.04	3.38 ± 0.93	0.457	2.55 ± 0.51	2.44 ± 0.46	0.847
subtotal	3.68 ± 0.85			3.45 ± 0.91			2.51 ± 0.49		
Amplitude (µV)	11.42 ± 4.66	11.64 ± 4.75	0.797	4.06 ± 1.82	3.93 ± 1.69	0.675	13.06 ± 5.86	11.88 ± 4.92	0.135
subtotal	11.53 ± 4.41			4 ± 1.58			12.64 ± 5.53		
Velocity (m/s)	3 9.47 ± 4.66	39.44 ± 5.99	0.973	42.73 ± 4.94	42.35 ± 4.84	0.669	58.04 ± 11.31	59.63 ± 13.08	0.613
subtotal	39.45 ± 4.68			42.54 ± 4.36			58.6 ± 11.86		

Data are mean ± SD; *p*-value for difference between the groups was calculated from the one-way ANOVA or Kruskal–Wallis test or chi-squared test.

**Table 3 jpm-13-00617-t003:** Mean values of the motor/the sensory conduction parameters of the tibial nerve and of the peroneal nerve by age group.

Index	Age (Years)				*p*-Value
	37−40 (*n* = 2)	41−50 (*n* = 7)	51−60 (*n* = 14)	> 60 (*n* = 38)	
The motor conduction of Tibial nerve					
Latency (ms)	3.1 ± 0.67	4.01 ± 1.10	3.62 ± 0.80	3.66 ± 0.83	0.608
Amplitude (µV)	15.52 ± 4.65	14.91 ± 4.77	11.79 ± 4.68	10.6 ± 3.96	0.871
Velocity (m/s)	44 ± 0.70	38.5 ± 3.98	40.53 ± 3.44	39 ± 5.18	0.358
The motor conduction of Peroneal nerve					
Latency (ms)	2.5 ± 0.14	4.18 ± 1.53	3.33 ± 0.74	3.41 ± 0.79	0.074
Amplitude (µV)	6.55 ± 0.21	3.78 ± 2.28	4.47 ± 1.74	3.72 ± 1.27	0.056
Velocity (m/s)	45.25 ± 0.35	39.35 ± 7.35	43.53 ± 4.43	42.62 ± 3.52	0.155
The sensory conduction					
Latency (ms)	2.03 ± 0.33	2.72 ± 0.7	2.61 ± 0.51	2.46 ± 0.44	0.280
Amplitude (µV)	18.97 ± 13.96	11.62 ± 5.41	13.57 ± 6.21	12.08 ± 4.71	0.325
Velocity (m/s)	77 ± 21.21	52.5 ± 11.64	56.15 ± 8.63	59.56 ± 11.77	0.062

Data are mean ± SD; *p*-value for difference between the groups was calculated from the one-way. ANOVA or Kruskal–Wallis test or chi-squared test.

**Table 4 jpm-13-00617-t004:** Mean values of the motor/the sensory conduction parameters of the tibial nerve and the peroneal nerve by gender.

Index	Tibial Nerve Conduction			Peroneal Nerve Conduction			Sensory Conduction		
	Male (*n* = 39)	Female (*n* = 22)	*p*-Value	Male (*n* = 39)	Female (*n* = 22)	*p*-Value	Male (*n* = 35)	Female (*n* = 19)	*p*-Value
Latency (ms)	3.89 ± 0.82	3.3 ± 0.78	0.085	3.6 ± 1.01	3.16 ± 0.62	0.076	2.55 ± 0.51	2.44 ± 0.46	0.446
Subtotal	3.68 ± 0.85			3.45 ± 0.91			2.51 ± 0.49		
Amplitude (µV)	11.68 ± 3.61	11.25 ± 5.65	0.749	3.93 ± 1.68	4.12 ± 1.43	0.658	13.06 ± 5.86	11.88 ± 4.92	0.461
Subtotal	11.53 ± 4.41			4 ± 1.58			12.64 ± 5.53		
Velocity (m/s)	39.37 ± 4.12	39.61 ± 5.65	0.861	41.78 ± 4.39	43.95 ± 4.05	0.066	58.04 ± 11.31	59.63 ± 13.08	0.643
Subtotal	39.45 ± 4.68			42.54 ± 4.36			58.6 ± 11.86		

Data are mean ± SD; *p*-value for difference between the groups was calculated from the one-way ANOVA or Kruskal–Wallis test or chi-squared test.

**Table 5 jpm-13-00617-t005:** The relationship between clinical symptoms and duration of diabetes.

Clinical Symptoms	Overall	Duration of Diabetes			
		<5 Years	5–10 Years	>10 Years	*p*-Value
Paresthesia	data	data			
No symptoms	14 (23.00)	13 (21.31)	1(1.63)	0	<0.001
Numbness	33 (54.10)	6 (9.83)	15 (24.59)	12 (19.67)	<0.001
Burning sensation	5 (8.20)	0	0	5 (8.19)	<0.001
Crawling ants	28 (45.90)	9 (14.75)	11 (18.03)	8 (13.11)	<0.001
Loss of sensation	2 (3.30)	1 (1.63)	0	1 (1.63)	-
Rough touch					
Normal	29 (47.54)	22 (36.06)	7 (11.47)	0	<0.001
Decreased	32 (52.56)	5 (8.19)	14 (22.95)	13 (21.31)	<0.001
Loss	0	0	0	0	-
Deep sensation					
Normal	57 (93.44)	26 (42.62)	21 (34.42)	10 (16.39)	<0.001
Decreased	4 (6.56)	1 (1.63)	0	3 (4.91)	-
Loss	0	0	0	0	-
Movement disorder					
Normal	58 (95.08)	27 (44.26)	20 (32.78)	11 (18.03)	<0.001
Weak	3 (4.92)	0	1 (1.63)	2 (3.27)	-

**Table 6 jpm-13-00617-t006:** Relationship between peripheral neuropathy and diabetes duration/group of BMI.

Nerve Damage	Overall	Duration of Diabetes				BMI ^1^ (kg/m^2^)			
		<5 Years	5−10 Years	>10 Years	*p*-Value	<18.5	18.5−22.9	≥23	*p*-Value
Right tibial	41 (67.21)	19 (31.14)	7 (11.47)	15 (24.59)	0.284	3 (4.91)	20 (32.78)	20 (32.78)	0.447
Left tibial	42 (68.85)	18 (29.50)	9 (14.75)	15 (24.59)	0.858	3 (4.91)	23 (37.70)	23 (37.70)	0.296
Right peroneal	52 (86.70)	23 (38.33)	10 (16.66)	19 (31.66)	0.849	3 (5.00)	29 (48.33)	29 (48.33)	0.145
Left peroneal	52 (86.70)	23 (38.33)	10 (16.66)	19 (31.66)	0.849	4 (6.66)	28 (46.66)	28 (46.66)	0.145
Right superficial peroneal	12 (19.70)	4 (7.01)	2 (3.50)	6 (10.52)	0.628	1 (1.75)	5 (8.77)	5 (8.77)	0.476
Left superficial peroneal	10 (17.20)	4 (6.89)	2 (3.44)	4 (6.89)	0.917	2 (3.44)	6 (10.34)	6 (10.34)	0.053

^1^ Abbreviation: BMI, body mass index.

**Table 7 jpm-13-00617-t007:** Relationship between the index of tibial nerve, the peroneal nerve, and the superficial peroneal nerve with HbA1c and hypertension.

	Neurotransmitter Index	HbA1c			Hypertension		
		≤7.5%	>7.5%	*p*-Value	Yes	No	*p*-Value
Tibial nerve							
Right	Latency	3.32 ± 0.69	3.86 ± 1.00	0.098	3.79 ± 0.99	3.73 ± 0.97	0.040
	Amplitude	9.70 ± 4.29	11.80 ± 4.70	0.179	11.61 ± 4.74	11.23 ± 4.66	0.259
	Velocity	39.00 ± 4.47	39.58 ± 4.74	0.712	40.03 ± 4.76	38.93 ± 4.57	0.380
	Latency	3.19 ± 0.84	3.68 ± 0.98	0.134	3.71 ± 1.04	3.48 ± 0.90	0.144
Left	Amplitude	9.48 ± 3.97	12.11 ± 4.81	0.096	11.67 ± 4.97	11.61 ± 4.61	0.690
	Velocity	40.45 ± 8.12	39.22 ± 5.49	0.541	39.70 ± 5.63	39.19 ± 6.19	0.108
Peroneal nerve							
R Right	Latency	3.12 ± 0.56	3.61 ± 1.11	0.162	3.69 ± 1.13	3.34 ± 0.93	0.020
	Amplitude	4.60 ± 2.04	3.94 ± 1.76	0.288	3.83 ± 1.81	4.30 ± 1.82	0.441
	Velocity	45.09 ± 5.02	42.20 ± 4.81	0.080	42.46 ± 5.35	43.00 ± 4.57	0.024
	Latency	3.04 ± 0.59	3.46 ± 0.98	0.176	3.48 ± 1.00	3.28 ± 0.86	0.003
Left	Amplitude	4.18 ± 2.15	3.87 ± 1.59	0.595	3.85 ± 1.70	4.01 ± 1.70	0.741
	Velocity	44.18 ± 5.58	41.93 ± 4.63	0.168	42.33 ± 5.31	42.36 ± 4.42	0.463
Superficial peroneal nerve							
Right	Latency	2.32 ± 0.59	2.58 ± 0.74	0.311	2.56 ± 0.82	2.50 ± 0.60	0.823
	Amplitude	12.93 ± 6.07	11.25 ± 5.50	0.393	12.53 ± 5.70	10.52 ± 5.36	0.554
	Velocity	64.80 ± 21.21	57.80 ± 16.87	0.260	59.20 ± 17.81	58.85 ± 17.91	0.985
	Latency	2.46 ± 0.50	2.52 ± 0.56	0.770	2.51 ± 0.53	2.51 ± 0.57	0.619
Left	Amplitude	12.86 ± 7.79	13.40 ± 6.81	0.822	13.98 ± 6.86	12.69 ± 7.04	0.734
	Velocity	59.10 ± 12.39	57.27 ± 12.64	0.678	57.25 ± 11.33	57.90 ± 13.71	0.598

**Table 8 jpm-13-00617-t008:** Relationship between the rate of nerve damage with HbA1c, hypertension, and kidney function.

Nerves	HbA1C			Hypertension			Dyslipidemia			Kidney Failure		
	≤7.5%	>7.5%	*p*-Value	Yes	No	*p*-Value	Yes	No	*p*-Value	Yes	No	*p*-Value
Right tibial	13.11 (8)	54.09 (33)	0.000	34.42 (21)	32.78 (20)	0.876	45.90 (21)	21.31 (20)	0.879	63.93 (39)	3.27 (2)	0.000
Left tibial	14.75 (9)	54.09 (33)	0.000	34.42 (21)	34.42 (21)	1.000	37.70 (23)	31.15 (19)	0.649	67.21 (41)	1.63 (1)	0.000
Right peroneal	13.33 (8)	73.33 (44)	0.000	41.66 (25)	45.00 (27)	0.782	47.54 (29)	37.70 (23)	0.721	85.00 (51)	1.66 (1)	0.000
Left peroneal	15.00 (9)	71.66 (43)	0.000	43.33 (26)	43.33 (26)	1.000	45.90 (28)	39.34 (24)	0.261	85.00 (51)	3.33 (2)	0.000
Right superficial peroneal	1.75 (1)	19.29 (11)	0.004	14.03 (8)	7.01 (4)	0.248	11.48 (7)	8.20 (5)	0.863	21.05 (12)	0.00 (0)	0.000
Left superficial peroneal	1.72 (1)	15.51 (9)	0.011	10.34 (8)	6.89 (4)	0.527	11.48 (7)	4.92 (3)	0.493	17.24 (10)	0.00 (0)	0.000

## Data Availability

The data used to support the findings of this study are available from the corresponding author upon request.

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
