# Peer review of "Lower Extremity Nerve Conduction Abnormalities in Vietnamese Patients with Type 2 Diabetes: A Cross-Sectional Study on Peripheral Neuropathy and Its Correlation with Glycemic Control and Renal Function"

_jpm, 2023, doi:10.3390/jpm13040617_

Round 1

Reviewer 1 Report

This study examined the conduction parameters of tibial nerve and peroneal nerve in type 2 diabetes and studied the correlations with other risk factors of diabetes. I believe the results are clinically useful because only a handful of studies were carried out in Southeast Asian populations. So I think it will also be useful for other countries as well and definitely not just Vietnam. The conclusion is well supported by the results. There are a few comments to improve the paper. 

How many examiners were used to person the neurological examination? How many times? What are their expertise and/or experience? Were they blinded to the clinical data? In my experience, the results of these tests might be subjective and biases might have been introduced. Inter-rater reliability test is also needed. Please explain. 

The authors should be aware of a recent meta-analysis which analyzed the tibial nerve cross-sectional area measured using ultrasonography in diabetic patients and correlated it with other factors such as BMI, peripheral neuropathy status. Please include the finding of this study in the discussion: https://www.mdpi.com/1648-9144/58/12/1696

Line 53: please combine refs #2 and #3 into a single bracket. This also applies to line 61, 64 and throughout the paper. 

First sentence in section 4.2 should be removed.

The sentence in line 262-263 needs a reference.

Please add the ethical approval number to the methods and add “Institutional Review Board Statement” after the conclusion (according to MDPI guidelines).

Author Response

Thank you for your valuable comments and suggestions. We have carefully addressed all the comments in the file below and made changes in the revised manuscript. All the changes are highlighted in yellow color.

Reviewer 1

This study examined the conduction parameters of tibial nerve and peroneal nerve in type 2 diabetes and studied the correlations with other risk factors of diabetes. I believe the results are clinically useful because only a handful of studies were carried out in Southeast Asian populations. So I think it will also be useful for other countries as well and definitely not just Vietnam. The conclusion is well supported by the results. There are a few comments to improve the paper. 

  1. How many examiners were used to person the neurological examination? How many times? What are their expertise and/or experience? Were they blinded to the clinical data? In my experience, the results of these tests might be subjective and biases might have been introduced. Inter-rater reliability test is also needed. Please explain. 

Answer: The nerve conduction recordings were conducted by two teams comprising of masters with a minimum of 8 years of experience in neurology and familiarity with nerve conduction meters, as well as one trained technician who had been involved in the measurement process for 6-7 years. The measurement process was performed in a completely blinded manner, where the physicians and technicians involved were unaware whether the nerve conduction measurements were for the subjects in this study or for inpatients receiving diabetes treatment or outpatients undergoing regular check-ups. Patients were assigned to the measurement room along with other common patients, including those who were not diagnosed with diabetes, and were referred by multiple doctors from the hospital.

  1. The authors should be aware of a recent meta-analysis which analyzed the tibial nerve cross-sectional area measured using ultrasonography in diabetic patients and correlated it with other factors such as BMI, peripheral neuropathy status. Please include the finding of this study in the discussion: https://www.mdpi.com/1648-9144/58/12/1696

Answer: We have added the findings of the following research into the discussion as followed:

A recent Systematic Review and Meta-Analysis of Ultrasonography Studies indicated that there was a weakly positive trend between age and tibial nerve CSA for both diabetic patients (r = 0.35, p = 0.24) and diabetic patients with DPN (r = 0.27, p = 0.34), it was not statistically significant.

Quote: Senarai, T., Pratipanawatr, T., Yurasakpong, L., Kruepunga, N., Limwachiranon, et al. (2022). Cross-Sectional Area of the Tibial Nerve in Diabetic Peripheral Neuropathy Patients: A Systematic Review and Meta-Analysis of Ultrasonography Studies. Medicina58(12), 1696.

  1. Line 53: please combine refs #2 and #3 into a single bracket. This also applies to line 61, 64 and throughout the paper. 

Answer: References have been combined into single brackets.

  1. First sentence in section 4.2 should be removed.

Answer: First sentence in section 4.2 has been removed

  1. The sentence in line 262-263 needs a reference.

Answer: Reference has been added to the sentence in line 262-263

  1. Please add the ethical approval number to the methods and add “Institutional Review Board Statement” after the conclusion (according to MDPI guidelines).

Answer: The ethical approval number and “Institutional Review Board Statement” has been added

Reviewer 2 Report

This is a cross sectional study investigating peripheral neuropathy in Vietnamese patients with type 2 diabetes. The main finding was a correlation between neuropathy and diabetes duration and glycemic control.

The topic of the study is quite interesting since diabetic neuropathy is the diabetes complication with poorer evidences. However my concerns regard mainly the very little population and the study design. 

Following some comments:

Introduction

This section is poor. Authors should highlight that diabetes complication (in particular Macrovascular) are presents also in prediabetes status. Accordingly, some studies should be cited: PMID: 34625357. Furthermore it is necessary to pay attention on potential target organ not usually associated with diabetes such as periodontitis (PMID: 35913467).

The authors wrote "this study aims to investigate 66 parameters of nerve conduction and the relationship between these parameters in lower 67 extremities and risk factors in type 2 diabetes patients in Vietnam.”. This statement is too generic. It is necessay to be more specific on what parameters Authors are focusing their study.

Methods

This section should be improved.

Can you give some detailes regarding recruitment modalities and the characteristics of the center? Is it an outward clinic of second level?

Statistical analysis

please, explain how sample size was calculated and what was the primary outcome.

Results

Table 1: the title of the table “this is a table etc…”) should be avoidedi. Furthermore, it is not acceptable to write p>0.05 or p<0.05. Exact p values should be provided for all the variables.

i noted that patients have a very bad glycemic control: more than 10%. This is quite surprising. Can you provide a reason?

this is quite curious expecially considering the high number of patients with low diabetes duration and the low BMI of the studied population. (opportunistic recruitment?)

Discussion

the discussion is very long and not focused. Please avoid results repetition and focus only on you main results underlying the difference and analogy with other similar studies.

Author Response

Thank you for your valuable comments and suggestions. We have carefully addressed all the comments in the file below and made changes in the revised manuscript. All the changes are highlighted in yellow color.

Reviewer 2

This is a cross sectional study investigating peripheral neuropathy in Vietnamese patients with type 2 diabetes. The main finding was a correlation between neuropathy and diabetes duration and glycemic control.

The topic of the study is quite interesting since diabetic neuropathy is the diabetes complication with poorer evidences. However, my concerns regard mainly the very little population and the study design. 

Following some comments:

Introduction

  1. This section is poor. Authors should highlight that diabetes complication (in particular Macrovascular) are presents also in prediabetes status. Accordingly, some studies should be cited: PMID:  Furthermore, it is necessary to pay attention on potential target organ not usually associated with diabetes such as periodontitis (PMID: 35913467).

Answer: We have added the findings of the following research into the introduction:

In contrast to neuropathy, pre-diabetes is often associated with early cardiovascular and kidney diseases, indicated by thickening of the endocardium and elevated glomerular filtration rate due to insulin resistance. However, the relationship between cardiovascular and renal complications is seldom discussed.

Di Pino A, Scicali R, Marchisello S, Zanoli L, Ferrara V, Urbano F, Filippello A, Di Mauro S, Scamporrino A, Piro S, Castellino P, Purrello F, Rabuazzo AM. High glomerular filtration rate is associated with impaired arterial stiffness and subendocardial viability ratio in prediabetic subjects. Nutr Metab Cardiovasc Dis 2021;31:3393-400.

The severity of the disease increases with disease duration and is often accompanied by an increase in complications such as cardiovascular disease, nephropathy, eye complications, foot complications, and periodontitis.

Nibali L, Gkranias N, Mainas G, Di Pino A. Periodontitis and implant complications in diabetes. Periodontol 2000 2022;90:88-105.

  1. The authors wrote "this study aims to investigate 66 parameters of nerve conduction and the relationship between these parameters in lower 67 extremities and risk factors in type 2 diabetes patients in Vietnam.”. This statement is too generic. It is necessary to be more specific on what parameters Authors are focusing their study.

Answer: Thank you for your comment. We have re-written this sentence as “Therefore, this study aims to investigate parameters of nerve conduction (e.g. peripheral motor potential time, response amplitude M, and motor conduction speed, as well as sensory conduction in the shallow nerve), as well as the relationship between these pa-rameters in lower extremities and risk factors in type 2 diabetes patients in Vietnam” to specify the focus of the our study.

Methods

This section should be improved.

  1. Can you give some details regarding recruitment modalities and the characteristics of the center? Is it an outward clinic of second level?

Answer: The study was conducted at a specialized General Hospital affiliated with a prominent medical university in Vietnam, which boasts of 36 departments including the Department of Endocrinology for treating diabetes patients and the Department of Neurology. This facility is recognized as a prestigious center for medical examination, treatment, research, and training. The website of the affiliated School-Hospital can be accessed for further information:  http://vmmu.edu.vn/Portal/Home.html; http://www.benhvien103.vn/

Statistical analysis

  1. Please explain how sample size was calculated and what was the primary outcome.

Answer: The sample size of 61 patients was determined based on practical considerations, specifically the availability of patients who met the inclusion criteria within a period of 9 months. While this sample size may appear small based on statistical considerations, appropriate statistical methods were used to analyze the data, allowing for meaningful conclusions to be drawn from this sample size.

The primary outcome of the study highlights:

  • The high incidence of peripheral neuropathy among T2DM patients in Vietnam
  • The association between peripheral nerve damage and risk factors such as, blood sugar control, kidney damage, BMI, age, and duration of diabetes.
  • The association between nerve conduction abnormalities and poor glucose control and/or decreased renal function

Results

  1. Table 1: the title of the table “this is a table….”) should be avoided. Furthermore, it is not acceptable to write p>0.05 or p<0.05. Exact p values should be provided for all the variables.

Answer: The table's title has been revised to avoid using the phrase 'This is a table...'. Additionally, we have corrected the presentation of p-values by providing the exact values for all variables rather than using p<0.05 or p>0.05

  1. I noted that patients have a very bad glycemic control: more than 10%. This is quite surprising. Can you provide a reason? This is quite curious especially considering the high number of patients with low diabetes duration and the low BMI of the studied population. (opportunistic recruitment?)

Answer: The study period coincided with the COVID epidemic, so the patient did not go to the doctor regularly; By the time you get to the doctor, it's too late, potentially requiring hospitalization due to high blood sugar. During the epidemic, the re-examination took place sporadically and regularly. In addition, the routine check-up interval is extended by 3 months, instead of the usual 1 month.

All patients who meet the selection criteria are ensured to be unbiased during the sample collection process. In Vietnam and some Southeast Asian countries, the majority of diabetes patients have a low BMI or are just overweight and are not obese. This contrasts Europe and America, where diabetes rates are higher among overweight and obese individuals. However, diabetes in Vietnam is characterized by a higher rate of abdominal obesity and visceral fat, which distinguishes it from other populations.

Discussion

  1. The discussion is very long and not focused. Please avoid results repetition and focus only on your main results underlying the difference and analogy with other similar studies.

Answer: The discussion has been shortened and focused on key results, all repetition has been removed

Round 2

Reviewer 1 Report

I thank the authors for the revised version. I have read the second version and I have no further comments. Congratulations!

Reviewer 2 Report

No further comments needed